# Melphalan, Etoposide, and Carboplatin Megatherapy with Autologous Stem Cell Transplantation in Children with Relapsing or Therapy-Resistant Extracranial Germ-Cell Tumors—A Retrospective Analysis

**DOI:** 10.3390/cancers12123841

**Published:** 2020-12-19

**Authors:** Marek Ussowicz, Monika Mielcarek-Siedziuk, Jakub Musiał, Mateusz Stachowiak, Jadwiga Węcławek-Tompol, Dorota Sęga-Pondel, Jowita Frączkiewicz, Joanna Trelińska, Anna Raciborska

**Affiliations:** 1Department of Paediatric Bone Marrow Transplantation, Oncology and Hematology, Wrocław Medical University, 50-367 Wrocław, Poland; mmielcarek@gmail.com (M.M.-S.); mateuszjulianstachowiak@gmail.com (M.S.); jadwiga.weclawek@gmail.com (J.W.-T.); dsega@poczta.onet.pl (D.S.-P.); jowitafr@gmail.com (J.F.); 2Department of Pediatric Oncology and Hematology, Clinical Hospital No. 2, 35-301 Rzeszów, Poland; jakubjmusial@gmail.com; 3Department of Pediatrics, Oncology, Hematology and Diabetology, Medical University of Lodz, 91-738 Łódź, Poland; joanna.trelinska@umed.lodz.pl; 4Department of Oncology and Surgical Oncology for Children and Youth, Institute of Mother and Child, 01-211 Warsaw, Poland; anna.raciborska@hoga.pl

**Keywords:** germ cell tumors, high dose chemotherapy, autologous stem cell transplantation

## Abstract

**Simple Summary:**

Germ cell tumors (GCTs) are malignancies derived from germ cells that originate in gonads or extragonadal localizations. They are considered highly curable in both children and adults even if distant metastases are present, but therapy-resistant or relapsing patients have a worse prognosis. The aim of our retrospective study was to analyze the outcome of 18 children with GCT treated with melphalan–etoposide–carboplatin high-dose chemotherapy and autologous stem cell transplantation. To date, this is one of the largest reported pediatric cohorts of GCT patients treated with megatherapy. We observed high survival rates—a five-year overall survival of 76%, and event-free survival of 70.8% without therapy-associated mortality. We concluded that this megatherapy protocol is feasible in heavily pretreated children, but the issue of precise indications for high dose chemotherapy (HDCT) is evident and must be answered in a well-designed controlled study to avoid unnecessary overtreatment.

**Abstract:**

Pediatric germ cell tumors (GCTs) are a group of chemosensitive malignancies with a 90% curability rate. We report a series of children with relapsing or therapy-resistant GCT treated with melphalan–etoposide–carboplatin high-dose chemotherapy (HDCT) and autologous stem cell transplantation. This consisted of 18 children, either with GCTs after relapse (nine patients) or with an unsatisfactory response to first-line chemotherapy (nine patients), who underwent HDCT. The HDCT regimens MEC1 (carboplatin 1500 mg/m^2^, etoposide 1800 mg/m^2^, and melphalan 140 mg/m^2^) and MEC2 (carboplatin 800 mg/m^2^, etoposide 800 mg/m^2^, and melphalan 140 mg/m^2^) were each used in nine patients. The median observation time was 81 months, the 5-year overall survival (OS) was 76%, and the event-free survival (EFS) was 70.8%. Non-relapse mortality was 0%, and four patients died after HDCT due to progression of the malignancy. No difference in OS or EFS was noted between the MEC1 and MEC2 protocols. The 5-year OS and 5-year EFS were higher in children treated with autologous stem cell transplantation before the age of four years. The presence of metastatic disease or time of HDCT consolidation during first/subsequent line chemotherapy did not affect patient survival. The melphalan–etoposide–carboplatin protocol is feasible in pediatric GCT, but is associated with potentially life-threatening complications. In conclusion, the use of HDCT must be examined in well-designed clinical trials, and the identification of patients who can benefit from this approach is critical to avoid overtreatment.

## 1. Introduction

Germ cell tumors (GCTs) are malignancies derived from germ cells that originate in gonads or extragonadal localizations. GCT are divided into two main categories: germinomatous or seminomatous germ cell tumors (SGCT), and nongerminomatous or nonseminomatous germ cell tumors (NSGCT). The SGCT group consists of germinomas, dysgerminomas, and seminomas, and show high chemo- and radiosensitivity. The NSGCT group includes other subtypes, such as teratomas, yolk sac tumors, embryonal carcinomas, choriocarcinomas, polyembryomas, and gonadoblastomas, and are characterized by aggressive growth and high chemosensitivity to platinum compounds. GCT are considered as highly curable tumors in both children and adults, even if distant metastases are present, but therapy-resistant or relapsing patients have a worse prognosis. According to the International Germ-Cell Cancer Cooperative Group, the primary tumor site, the response to first-line treatment, the progression-free interval between first-line therapy and relapse, tumor markers at relapse, and the presence of liver, bone, or brain metastases at relapse are associated with curability [1]. In adults, high-dose chemotherapy (HDCT) followed by autologous stem-cell transplantation (ASCT) has been introduced into the treatment of progressing or relapsing tumors, with a curability rate of 60% [2,3]. The HDCT backbone in adults is usually based on a carboplatin–etoposide combination, but there are no established protocols in the pediatric population. The purpose of this analysis was to determine the feasibility of melphalan, etoposide, and carboplatin HDCT with ASCT in children with recurrent or therapy-resistant GCT.

## 2. Results

### 2.1. Patient and Chemotherapy Characteristics

The analyzed cohort consisted of 18 children with malignant GCT who were referred for HDCT with ASCT. Referral was based on the individual analysis of clinical response, and only children with advanced malignant tumors, with multiple relapses, or with the presence of metastatic disease were treated with HDCT. Referred patients qualified for the HDCT procedure after consultation with the national coordinating center, as HDCT was never a part of the standard first-line therapy. Baseline patient and treatment characteristics are summarized in Table 1. 

In the studied group, 12 patients were diagnosed with extragonadal disease and 6 with gonadal tumors. In 11 patients, metastases were present at initial diagnosis. Surgery was performed at different points of therapy, and the extent of the resection ranged from biopsy to complete resection. First-line therapy included chemotherapy based on the TGM-95 study, with surgical resection of primary tumors. In the first line, patients were treated with vinblastine/bleomycin/cisplatin chemotherapy (VBP, vinblastine 6 mg/m^2^, bleomycin 45,000 IE/m^2^, cisplatin 100 mg/m^2^) and/or vinblastine/bleomycin/cisplatin chemotherapy (VIP, cisplatin 100 mg/m^2^, ifosfamide 6 g/m^2^, etoposide 375 mg/m^2^). Patients treated with first-line therapy were classified as high-risk if initial alpha-fetoprotein (AFP) levels were high (> 15,000 ng/mL) or disseminated disease was found. Three patients (unique patient number (UPN) 3, 5, and 9) were initially treated as standard risk patients (VBP protocol), but after relapse or due to advanced disease, were treated with VIP and doxorubicin/bleomycin/carboplatin chemotherapy (ABK) chemotherapy. Two children received one VBP cycle before staging was complete, but were then classified as high risk, and the remaining 13 patients were stratified upfront as high-risk due to a high initial AFP concentration. Lack of AFP normalization after three VBP cycles or four VIP cycles was a criterium of unfavorable response, and was used as an indicator to proceed with next line of chemotherapy (VIP or ABK, respectively). After VIP or VBP + VIP chemotherapy in nine patients, their responses did not fulfill the complete remission (CR) criteria (both radiological remission and normal AFP level), and nine patients experienced one or multiple relapses of disease. In 10 patients, the disease was disseminated at the time of autologous stem cell transplantation, and eight showed poor local control of the tumor. After first-line therapy failure, patients were treated with various chemotherapy protocols. In 13 patients, the ABK protocol (bleomycin 45,000 IE/m^2^, carboplatin 600 mg/m^2^, doxorubicin 60 mg/m^2^) was administered. One patient received the BEP protocol (bleomycin 30,000 IE/m^2^, etoposide 500 mg/m^2^, cisplatin 100 mg/m^2^). In two patients, stem cell apheresis was performed after an additional mobilization protocol consisting of cyclophosphamide at a dose of 4 g/m^2^ (HDCY). None of the treated patients underwent radiotherapy. Patients were grouped according to their response to first-line chemotherapy as patients with primary refractory or persistent disease or as relapsed patients who had disease recurrence at any time prior to HDCT referral.

### 2.2. High-Dose Chemotherapy with Autologous Stem Cell Transplantation

HDCT data are presented in Table 2. Peripheral blood stem cell (PBSC) collection was performed after a chemotherapy cycle in 17 patients (UPN 1–17) and in two patients, bone marrow (BM) harvest was necessary (UPN 15 and UPN 18). The original HDCT protocol, MEC1, given in years 2003–2010 to nine patients (UPN 10–18), consisted of melphalan at a dose of 140 mg/m^2^ on day −6, etoposide at a dose of 1800 mg/m^2^ on day −5, and carboplatin at a dose of 500 mg/m^2^/day on days −4 to −2.

Due to severe mucositis with life threatening bleeding or sepsis following the MEC1 protocol (in patients UPN 13 and UPN 15), the decision to reduce the dose of HDCT was made in 2011. The second HDCT protocol, MEC2, administered after 2011 in nine patients (UPN 1–9), consisted of carboplatin at a dose of 200 mg/m^2^/day on days −6 to −3, etoposide at a dose of 200 mg/m^2^/day on days −6 to −3, and melphalan at a dose of 140 mg/m^2^ on day −1. The median age at HDCT was 40 months (range 17–222). The graft material contained a median of 4.56 × 10^6^ CD34 cells/kg (range 1.25–13.22 × 10^6^ CD34 cells/kg). All patients engrafted and achieved trilineage bone marrow recovery.

### 2.3. Survival Analysis

Non-relapse mortality (NRM) in the whole group was 0%, and four patients died after ASCT due to progression of the malignancy. Reversible toxicities assessed according to the Common Terminology Criteria for Adverse Events (CTCAE) were observed in all patients [4]. All 18 patients showed grade 4 leukopenia and neutropenia. Children exhibited fever of unknown origin grade 3 (88%) or 4 (11%), and mucositis grade 3 (88%) or 4 (11%). In four patients, sepsis grade 3 or 4 was diagnosed, and bacteremia grade 2 was diagnosed in one patient. Hypertransaminasemia was found in 11 patients (61%), grade 1–3 in 10 children, and grade 4 in one patient who developed hepatic veno-occlusive disease. In two patients, the maximum creatinine concentration was above the normal range, and these patients showed acute kidney injury but did not require hemodialysis.

Transient acute renal failure was present in one patient (UPN 11). After discharge, patients were followed up at the transplant center or at primary oncological centers for a median of 81 months (range 0.8 to 181 months). Residual tumor biopsies were performed in three children after HDCT, and viable tumor cells were found in two patients showing radiological progression with elevated AFP. Due to the long observation period, two late events were recorded—in patient UPN 16, a local relapse was observed 16 years after HDCT, and in patient UPN 14, osteosarcoma was diagnosed 12 years after HDCT—that affected the right-hand part of the survival curve. The detailed results of the survival analysis are shown in Table 3.

The 5-year OS and 5-year EFS of the entire cohort was 76% and 70.8%, respectively (Figure 1A). The comparison between survival after MEC1 and MEC2 HDCT showed no difference in OS or EFS (Figure 1B). Among the treated children were 10 boys and 8 girls, and these groups did not differ in terms of OS and EFS (Figure 1C). The analyzed cohort showed a bimodal age distribution, with the first group of patients diagnosed at an early age and transplanted before the age of four years (14 patients), and the second group of patients treated after the age of 12 years (five patients). The 5-year OS and 5-year EFS were superior in the younger cohort (Figure 1D). The seven patients undergoing HDCT as consolidation in first line chemotherapy, and the 11 children who were referred after one or more relapses, showed similar outcomes (Figure 1E). The presence of metastatic disease was diagnosed in 10/18 transplanted patients, but its impact on OS or EFS was not evident (Figure 1F).

## 3. Discussion

GCT subtypes and the therapeutic outcome in pediatric population are different from adult patients, and despite many similarities, the experience gained from large adult studies cannot be generalized. The efficacy of first-line chemotherapy, which was reported originally in the TGM-95 study, for children treated with this protocol in Poland is very high. Early experience with children treated for GCT showed a 14% first-line therapy failure rate, of which two-thirds of all patients died [5]. Overall survival in the high-risk group was 89%, while it reached 93% in the standard-risk group [6]. These outcome measures were similar to those reported by the original French study group [7]. The risk of treatment failure was similar in children with sacrococcygeal and gonadal localization [8]. Dysgerminoma patients had an excellent prognosis, even in advanced cases with conservative surgery and platinum-based chemotherapy [9].

Only a minority of GCT patients develop highly resistant disease or multiple recurrences. However, therapy for these cases is not clearly outlined in the pediatric population due to the rarity of this condition. In adults, 40–50% of relapsing patients may reach long-term remission after salvage chemotherapy, such as TIP (paclitaxel, ifosfamide, and cisplatin) or VeIP (vinblastine, ifosfamide, cisplatin) [10]. The transplant strategy in adults was reviewed by bin Riaz et al., who concluded that a single cycle of HDCT with ASCT does not improve outcomes, but two or three cycles of HDCT improved survival for patients with refractory or relapsed GCT [11]. Early studies of consolidation therapy with HDCT containing carboplatin and etoposide with ASCT were associated with poor outcomes and 20% non-relapse mortality (NRM) [12]. The most effective HDCT protocols in adult studies were the Indiana regimen and the TI-CE protocol. The Indiana regimen consists of two consecutive cycles of HDCT (carboplatin 2100 mg/m^2^ and etoposide 2250 mg/m^2^), followed by ASCT, whereas the TI-CE regimen consists of three cycles of paclitaxel plus ifosfamide, followed by carboplatin plus etoposide HDCT with ASCT. Currently, a randomized trial (Alliance A031102 (TIGER trial)) is being conducted to compare standard dose chemotherapy vs. HDCT with ASCT in the adult population [13]. Notably, due to intensive chemotherapy protocols, the current adult studies on HDCT with ASCT report NRM rates of up to 7%.

In children, due to the rarity of the condition, the number of treated patients and published studies is very low, and the perspective of a randomized prospective study is unlikely. The largest group was reported by de Giorgi et al., who reviewed the records of the European Group for Blood and Marrow Transplantation (EBMT), and showed that 8 of 14 patients with extracranial GCTs remained in remission after HDCT. The HDCT protocols reported in the EBMT analysis were variable, with carboplatin–etoposide–cyclophosphamide (carboPEC), carboplatin–etoposide, and thiotepa–etoposide most commonly used. The French group reported that among the 273 children with non-seminomatous GCT in the TGM-95 study, treatment failure was initially observed in 19 patients (7%), and only 4 out of the 10 patients treated with HDCT survived [14]. The study reported 5-year EFS and OS of 26% and 32%, respectively. The conditioning in the French cohort was based on etoposide–thiotepa in five cases and carboPEC in four patients [15,16]. The Children’s Oncology Group (COG) group used a paclitaxel, ifosfamide, and carboplatin (TIC) regimen (paclitaxel 135 mg/m^2^/day on day 1, ifosfamide 1800 mg/m^2^/dose on days 1–5, and carboplatin with area under curve (AUC) 6.5 on day 1) as a second-line therapy in children with relapsed GCT and reported that the overall response rate was 44% using combined Response Evaluation Criteria in Solid Tumors (RECIST) response and tumor marker decline [17]. In this study, ACST was performed in five patients; however, only one of five patients was cured, and no details on the HDCT regimen were provided. In 2020, De Pasquale et al. published results from Italy, showing that in children with platinum-refractory GCT, OS is 54.5% (compared with 91.5% for the relapsed group), and a conditioning regimen based on thiotepa and cyclophosphamide was used in 15 of 16 patients [18]. In this study, 19 of 21 patients received a median of three cycles of the second line chemotherapy ICE (ifosfamide 1.8 g/m^2^ days 1 to 5, carboplatin 400 mg/m^2^ days 1 and 2, etoposide 100 mg/m^2^ days 1 to 5) and in nine patients, surgery was performed on the residual tumor mass.

We studied the addition of melphalan to the carboplatin–etoposide combination. Melphalan is a bifunctional alkylator that undergoes spontaneous hydrolysis in the plasma, with inactive monohydroxy- and dihydroxymetabolites appearing within minutes of drug administration [19]. About 15% of the drug is excreted intact in the urine. The role of pharmacokinetics and toxicities in patients with renal failure remains unclear. As reviewed by Shaw, renal function affects melphalan clearance, and pharmacokinetic interactions have been reported with carboplatin [20]. The pharmacokinetics of intravenous high-dose melphalan is mildly affected by impaired renal function, making it a useful therapeutic option in diseases with renal failure [21,22]. The feasibility and safety of the melphalan, etoposide, and carboplatin combination has been extensively studied. Melphalan was used for decades in HDCT for a wide number of lymphoid malignancies, such as multiple myeloma, non-Hodgkin lymphoma, and in pediatric solid tumors such as neuroblastoma and Ewing’s sarcoma. The protocols containing melphalan, etoposide, and carboplatin (MEC or CEM) have been intensively studied in neuroblastoma, but showed inferior efficacy to tandem transplantations with thiotepa–carboplatin + MEC or to single busulfan–melfalan (BuMel) HDCT, and higher toxicity in comparison to BuMel [23,24,25]. The efficacy of MEC HDCT for malignant rhabdoid tumors of the kidney was not proven, but this entity is associated with high therapy resistance and a very poor prognosis [26]. The role of MEC in brain tumors is limited and the protocol is usually included as a part of the tandem HDCT strategy [27]. MEC HDCT in osteosarcoma did not show any clear advantage, and the megatherapy strategy is considered ineffective in this indication [28]. Children with unfavorable nephroblastoma diagnoses tolerated MEC very well and good curability rates were observed [29,30,31]. The efficacy of melphalan HDCT in GCT has not been studied; only Nieto et al. reported concurrent bevacizumab and sequential HDCT using GemDMC (gemcitabine, docetaxel, melphalan, and carboplatin) and ICE (ifosfamide, carboplatin, and etoposide) in adult patients with poor-risk relapsed or refractory GCT [32]. The cumulative dose of melphalan in this study was 105 or 150 mg/m^2^, but the dose was split over three consecutive days.

The toxicity of MEC in all studies manifested as mucositis, gastrointestinal toxicity, and infections. However, one of the serious adverse effects observed in a more intensive CEM protocol was an overall incidence of 29% of transplant-associated thrombotic microangiopathy (TA-TMA) [33]. The risk of hepatic veno-occlusive disease (VOD) in neuroblastoma patients treated with CEM HDCT was 9% [24]. According to our study, the toxicities observed during the posttransplant period were manageable and not associated with mortality, but the MEC1 protocol was associated with significant toxicities in four out of nine children, and the decision regarding its discontinuation seems justified.

Detailed analyses of prognostic factors in the analyzed group showed only the age at HDCT as significant. The bimodal age distribution and better outcomes in younger children (<4 years) can be explained by the presence of therapy-resistant or relapsing immature teratomas that were typical in this subgroup and might have showed higher chemosensitivity and resectability.

The techniques leading to better local tumor control need to be emphasized in patients with GCT, because surgery is associated with improved survival [14,34,35]. In children with local recurrence of sacrococcygeal GCT, complete surgical resection represents the cornerstone of salvage treatment, and chemotherapy is limited to patients with inoperable or metastatic disease [34]. As demonstrated by Fizazi et al., a complete resection may be more critical than recourse to postoperative chemotherapy in the setting of post-chemotherapy viable malignant NSGCT, depending on the completeness of resection, International Germ Cell Consensus Classification (IGCCC) group, and percentage of viable cells. Postoperative chemotherapy did not appear to improve OS compared with surveillance and treatment only at relapse [36,37]. In addition, the presence of a teratoma, particularly a mature teratoma, in an NSGCT primary tumor is associated with a higher cumulative incidence of disease-related death, consistent with the hypothesis that differentiation is associated with adverse outcomes; this observation from the study by Hunt et al. emphasizes the curative role of surgery [38].

In our series, 13 of 18 patients were not eligible for complete resection at any stage of therapy before, but at HDCT, only 5 of 18 patients showed elevated AFP and 4 patients showed viable tumor cells in the tumor resected during or after standard dose chemotherapy.

Among the six patients (UPN 2, 4, 7, 8, 12, and 13) without metastatic disease at HDCT, three were not eligible for complete resection (UPN 2, 12, and 13), and the two who underwent ASCT after complete resection eventually relapsed. Among the seven post-pubertal patients, only three are alive (UPN 5, 11, and 18), and the best achievable resection prior to HDCT was not complete. The post-pubertal patients who died of disease (UPN 3, 6, 9, and 15) had inoperable, metastatic disease that progressed after HDCT. In one patient, the isolated local relapse after HDCT was successfully treated with surgery. Good control of the local tumor and metastatic disease prior to HDCT can be associated with better outcomes for treated patients.

Some effects may not have been identified in the reported group due to the low number of studied patients, but the survival analysis suggests the prognostic impact of metastatic disease and the intensity of HDCT (MEC1versus MEC2) warrant further attention.

During the retrospective analysis of our group, the issue of precise indications for megatherapy became evident, and the identification of patients who might benefit from HDCT without the unnecessary risk of excessive toxicities appears as a key question. To resolve this problem, a well-designed, controlled prospective clinical trial is needed, and HDCT must be cautiously studied in pediatric populations.

The use of HDCT in therapy-resistant pediatric GCT patients is only one of the current therapeutic strategies. No consensus has been achieved on the best HDCT protocol, and the ongoing TIGER trial compares the conventional dose TIP chemotherapy (paclitaxel, ifosfamide, and cisplatin) with a TI-CE HDCT arm (paclitaxel and ifosfamide, followed by high-dose carboplatin and etoposide) in patients with advanced GCT. Recent genomic discoveries have uncovered the molecular landscape of some pediatric GCT [39]. Malignant germ cell tumors have a low mutational burden, but the KIT, KRAS, TGF-BMP, and Wnt-catenin signaling pathway inhibitors and anti-CD30 immunotherapy can be used as targeted therapies in patients with resistant disease. An early study by Einhorn et al. did not demonstrate clinical efficacy for KIT inhibition with imatinib [40]. Programmed death ligand-1 (PD-L1) expression is also common in male germ cell tumors, and this subgroup could benefit from immunotherapeutic strategies using checkpoint blockade [41]. According to the study by Adra et al., checkpoint inhibitors did not appear to have clinically meaningful single-agent activity in refractory GCT [42].

## 4. Methods

### 4.1. Study Population

The patients analyzed in our study were referred from 3 large Polish pediatric oncology centers over a period of 15 years. There were 18 consecutive GCT patients transplanted in 2003–2018 at the Department of Pediatric Bone Marrow Transplantation, Oncology and Hematology in Wroclaw, who were retrospectively included in this study. Toxicity data from patients’ medical records during the procedure and 30 days after transplantation were graded in accordance with the Common Toxicity Criteria version 5 from the National Institute of Health [4]. Disease response was assessed using the RECIST criteria with the inclusion of tumor marker levels [43]. Complete response (CR) was defined as no measurable disease, and normal markers (βHCG and AFP). Partial response (PR) was defined as at least a 30% decrease in tumor burden, and for patients with ≥10-fold marker elevation, at least a 1 log (90%) reduction. Progressive disease (PD) was defined as an increase in tumor measurement by 20% or increase in markers by 1 log. Stable disease (SD) was defined as neither PR or PD. The parents gave their written informed consent for the treatment and analysis of their clinical data. Ethical approval was waived by the local Ethics Committee of the Wroclaw Medical University in view of the retrospective nature of the study, and all procedures performed were part of routine care.

### 4.2. Statistical Analysis

The end points were overall survival (OS), defined as the time from transplantation to death or the last report from patients with no event, and event-free survival (EFS), defined as the time from transplantation to progression, relapse, second malignancy, or death. Survival curves were estimated using the Kaplan–Meier method and compared with the log rank test. Statistical analysis and data presentation were performed with the computer software GraphPad Prism 6.07 (GraphPad Software, La Jolla, CA, USA) and Statistica 13.0 (Statsoft/Dell, Palo Alto, CA, USA).

## 5. Conclusions

Pediatric GCTs are highly curable cancers, even in patients who relapse, and standard dose chemotherapy with surgery are effective as both first-line treatment and as salvage therapies. The role of HDCT in GCT treatment is currently being evaluated and remains to be established in post-pubertal patients. Whether this experience is applicable to pediatric pre-pubertal patients requires further study.

In our study, heavily pretreated patients survived HDCT with ASCT and achieved solid survival rates, but our study is limited by the low number of patients and its retrospective design, with a high variability in histologic subtypes, different results of surgical treatment, and responses to standard dose chemotherapy.

The MEC HDCT toxicities in GCT patients were significant but manageable and did not result in treatment-associated mortality. A crucial problem is the identification of patients who could benefit from the intensification of therapy. Due to significant toxicities and the risk of long-term sequelae, HDCT should be only studied in well-designed and controlled prospective clinical trials.

## Figures and Tables

**Figure 1 cancers-12-03841-f001:**
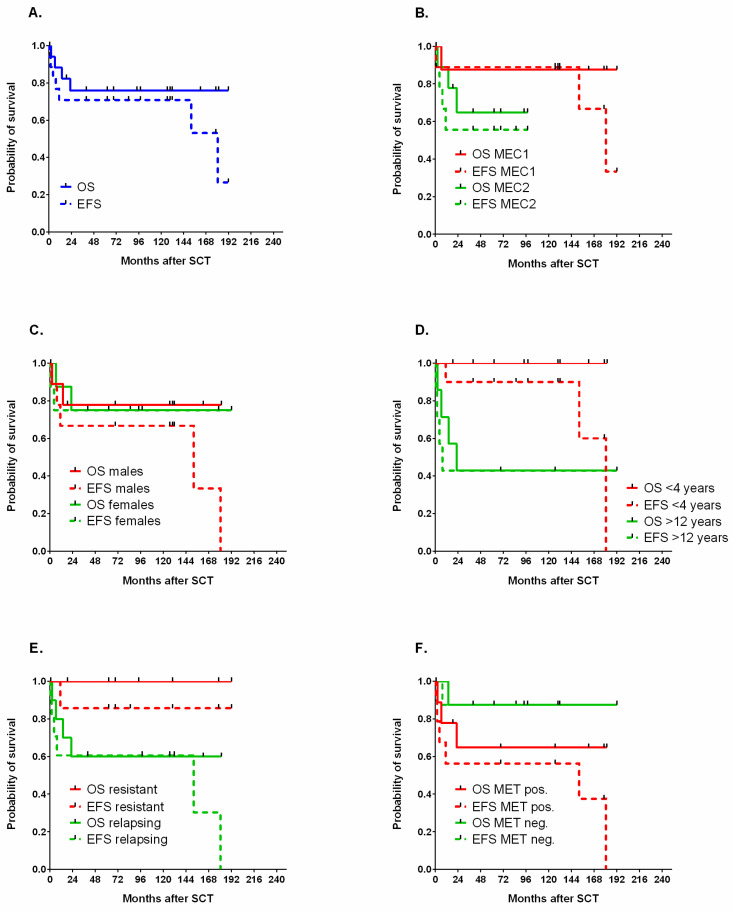
Probability of OS and EFS in the whole group. (**A**) In patients treated with MEC1 or MEC2 (**B**), male and female patients (**C**), children below the age of 4 years and above the age of 12 years (**D**), primary resistant or relapsing tumors (**E**), and in patients with disseminated disease (MET pos.) or a localized tumor (MET neg.) (**F**).

**Table 1 cancers-12-03841-t001:** Summary of demographics and baseline patient characteristics.

Patient Number (UPN)	Sex	Histology	Localization	First-Line Chemotherapy	Second Line Chemotherapy	Third Line Chemotherapy	Surgery	Indication for HDCT	Metastatic Disease at HDCT
1	M	GCT	extragonadal (sacrococcygeal), bone metastases	6 × VIP	4 × ABK		after 4 VIP, viable tumor (teratoma with blastemic fields) remained	advanced disease, viable tumor after 4 VIP chemotherapy	yes
2	F	GCT	extragonadal (sacrococcygeal)	6 × VIP	2 × ABK		1. after 4 VIP, 2. after 6 VIP	relapsed disease with residual tumor	no
3	F	YS	gonadal (ovary), pulmonary metastases	4 × VBP	4 × VIP	2 × ABK	3 times (resections and biopsies)	metastatic relapse	yes
4	F	YS	extragonadal (sacrococcygeal)	5 × VIP	3 × ABK		before chemotherapy	tumor rupture at surgery	no
5	M	GCT	gonadal (testis), abdominal lymph nodes and pulmonary metastases	4 × VIP	6 × ABK		before chemotherapy	advanced disease	yes
6	M	seminoma	gonadal (testis), abdominal lymph nodes, brain and pulmonary metastases	6 × VIP	4 × ABK		before HDCT	advanced metastatic disease	yes
7	F	YS/IT	extragonadal (sacrococcygeal)	5 × VIP	4 × ABK		before chemotherapy and at relapse	relapse after VIP chemotherapy	no
8	F	YS	extragonadal (sacrococcygeal), pulmonary metastases	6 × VIP	3 × ABK		1. after VIP, 2. prior to HDCT (no viable cells)	relapse after VIP chemotherapy	no
9	M	GCT	extragonadal (retroperitoneal)	4 × VBP	3 × VIP	4 × ABK	prior to HDCT	relapse after VIP chemotherapy	no
10	M	IT	extragonadal (sacrococcygeal), pulmonary metastases	missing data			local tumor prior to HDCT	metastatic relapse	yes
11	M	GCT	gonadal (testis)	6 × VIP	7 × BEP		local tumor prior to HDCT	advanced disease with metastatic pulmonary relapse	yes
12	M	YS/IT	extragonadal (mediastinal), pulmonary metastases	6 × VIP			before chemotherapy	advanced disease with unresectable tumor	no
13	M	YS	extragonadal (mediastinal tumor with extradural component in spinal canal)	missing data			before chemotherapy	relapsed disease after VIP chemotherapy	no
14	M	IT	extragonadal (sacrococcygeal), pulmonary and peritoneal metastases	1 × VBP	6 × VIP	3 × ABK	1. at diagnosis, 2. in relapse after VIP (viable cells)	advanced, metastatic disease, viable tumor after VIP chemotherapy	yes
15	F	embryonal carcinoma	gonadal (ovary)	4 × VIP	2 × ABK	PVB, POG/CCG 8882	before chemotherapy	metastatic peritoneal disease, multiple relapses	yes
16	M	IT	extragonadal (sacrococcygeal), pulmonary metastases	1 × VBP	5 × VIP	5 × ABK	multiple resections; the last—after 3 ABK, tumorectomy and metastasectomy, viable cells remained	disseminated disease with multiple relapses, viable tumor after 3 ABK chemotherapy	yes
17	F	embryonal carcinoma	extragonadal (sacrococcygeal), hepatic and pulmonary metastases	6 × VIP	ABK	HDCY	1. before chemotherapy, 2. after VIP chemotherapy	disseminated disease, viable tumor after VIP chemotherapy	yes
18	F	dysgerminoma	gonadal (ovary), mediastinal and brain metastases	6 × VIP	HDCY		1. before chemotherapy, 2. primary resection after 5 VIP	advanced metastatic disease	yes

ABK—doxorubicin/bleomycin/carboplatin chemotherapy, F—female, GCT—germ cell tumor (not otherwise specified), HDCT—high-dose chemotherapy, IT—immature teratoma, M—male, UPN—unique patient, VBP—vinblastine/bleomycin/cisplatin chemotherapy, VIP—etoposide/ifosfamide/cisplatin chemotherapy, and YS—yolk sac tumor number.

**Table 2 cancers-12-03841-t002:** HDCT characteristics, complications, and outcome.

Patient Number (UPN)	Age at HDCT in Months	Status at HDCT	AFP within 4 Weeks Prior to HDCT	HDCT Protocol	Graft Content 10^6^ CD34 Cells/kg	Mucositis, Grade	FUO/Infections	Toxicities with Grades	Post-HDCT Surgery	Post-HDCT Status	Follow-Up Time (Months)
1	39.0	1 PR	normal	MEC2	4.81	3	FUO 3	hepatic aminotransferase activity 2	after 13 months, mature teratoma	alive, local relapse	18.7
2	30.4	2 PR	n/a	MEC2	7.038	3	FUO 3	hepatic aminotransferase activity 2	no	A&W	40.0
3	184.7	2 PR	elevated	MEC2	4.83	3	FUO 3	hepatic aminotransferase activity 3	twice—debulking	DOD	22.7
4	41.7	1 PR	elevated	MEC2	7.86	3	FUO 3	hepatic aminotransferase activity 4, hepatic veno-occlusive disease 3	no	A&W	62.3
5	222.6	1 PR	normal	MEC2	11.1	3	FUO 3		no	A&W	69.1
6	212.9	3 PR	normal	MEC2	3.4	3	FUO 3	hepatic aminotransferase activity 1	no	DOD	2.4
7	28.6	1 CR	normal	MEC2	13.22	3	FUO 3	hepatic aminotransferase activity 2	no	A&W	94.1
8	26.6	2 CR	normal	MEC2	4.09	3	FUO 3	hepatic aminotransferase activity 1	no	A&W	97.8
9	209.2	2 PR	elevated	MEC2	4.48	3	FUO 3, sepsis 3		no	DOD	13.9
10	37.7	2 CR	normal	MEC1	3.52	3	FUO 3, sepsis 3		no	A&W	0.8
11	219.4	2 PR	normal	MEC1	3.93	3	FUO 3, bacteremia 2	hepatic aminotransferase activity 3, creatinine 3, hyponatremia 3, hyperkalemia 3, acute kidney injury 3	no	A&W	126.7
12	29.2	1 PR	normal	MEC1	3.12	3	FUO 3	hepatic aminotransferase activity 1	no	A&W	129.7
13	32.7	2 PR	elevated	MEC1	6.98	4	FUO 3, sepsis 4	mucosal and gastrointestinal bleeding 4	no	A&W	131.7
14	17.2	2 PR	normal	MEC1	5.11	3	FUO 3	hepatic aminotransferase activity 3	no	GCT remission, second malignancy (osteosarcoma)	162.2
15	176.8	2 PR	elevated	MEC1	1.25	4	FUO 3, sepsis 3	hepatic aminotransferase activity 2, creatinine 2, oliguria 2	no	DOD	6.3
16	26.3	3 CR	normal	MEC1	4.64	3	FUO 3		no	alive, local relapse	181.5
17	43.0	1 CR	normal	MEC1	2.84	3	FUO 3		no	A&W	178.6
18	147.3	1 PR	n/a	MEC1	2.4	3	FUO 3		yes, no malignancy	A&W	192.0

AFP—alpha-fetoprotein, A&W—alive and well, CR—complete remission, DOD—died of disease, FUO—fever of unknown origin, HDCT—high dose chemotherapy, n/a—not available, PR—partial remission, and UPN—unique patient number.

**Table 3 cancers-12-03841-t003:** Survival analysis results.

Category		Number of Patients	5 Year OS	Log Rank *p*	5 Year EFS	Log Rank *p*
All patients		18	76%		70.80%	
Type of HDCT	MEC1	9	87.50%	ns	88.90%	ns
	MEC2	9	64.80%		55.60%	
Sex	male	10	77.80%	ns	66.70%	ns
	female	8	75%		75%	
Age	<4 years	11	100%	0.007	90%	0.02
	>12 years	7	42.90%		43%	
Disease status at HDCT	1 CR/PR	7	100%	ns	85.7%	ns
	>1 CR/PR	11	60%		60.6%	
Metastatic disease	yes	10	64.80%	ns	56.30%	ns
	no	8	87.50%		87.50%	

CR—complete remission, EFS—event free survival, HDCT—high dose chemotherapy, n/a—non-applicable, ns—not significant, OS—overall survival, PR—partial remission.

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
