# Peer review of "Melphalan, Etoposide, and Carboplatin Megatherapy with Autologous Stem Cell Transplantation in Children with Relapsing or Therapy-Resistant Extracranial Germ-Cell Tumors—A Retrospective Analysis"

_cancers, 2020, doi:10.3390/cancers12123841_

Round 1
Reviewer 1 Report
The results of the work presented in this manuscript summarize novel, interesting and scientifically valuable informations. It seems to be a timely and useful contribution.
The manuscript is well written and provides new and interesting data supported by proper methods and data analyses.
It also provides some new and valuable insight into the existing therapy of germ cell tumors in children.
However, some minor corrections and additional data are required before accepting the manuscript.
The authors need to add some other previous or recent publications about the importance and potential value of other therapeutic options such as modern targeted therapeutic modalities, novel immunotherapies, hormonal treatments or studies with various kinase inhibitors. These therapeutic options are being evaluated for tumors including germ cell tumors that do not respond to chemotherapy.
Example: Maoz et al : Molecular Pathways and Targeted Therapies for Malignant Ovarian Germ Cell Tumors and Sex Cord-Stromal Tumors: A Contemporary Review. Cancers 2020.etc.
Reviewer 2 Report
Relevant topic, so thorough analysis of the data could be interesting. However, essential information is missing in the paper. It is unclear what the indications were for HDCT. Please give definitions of the situations in which HDCT was performed. Especially for young children with very good prognosis. Could you please specify in table 3 which patients had recurrence and which ‘unsatisfactory response’? Two patients with IT were also included. Did they experienced malignant relapse? What was the indication for transplant? No information is given on local therapy (surgery and/or radiotherapy) of the patients in the first line or subsequent therapies. Please give the details on this, including timing of the local Tx. Please provide more information on the initial patient and tumor status: stage, site, risk category (with definition / according to which system). Authors state that patients in first line treatment were high risk based on high AFP and/or delayed response. Please provide definitions and reference for this classification. Please give definitions of CR and PR, what does it mean that patient do not fulfill CR criteria. What were the criteria? Was there any biopsy of residual disease done? With which results? To assess feasibility of the regimen more data on toxicity is needed. In the paper is only mentioned that main toxicity was mucositis and myelosuppression. Please provide number of patients with grade of toxicities. As authors states, after MEC1 was severe mucositis; what does that mean? Is that regimen still feasible/ acceptable? For patients above 4 yrs the survival remained unsatisfactory. It seems that HDCT did not help. For the younger children is the question whether they really needed HDCT. Could you please comment on these remarks?
Reviewer 3 Report
Authors report a retrospective study conducted on the cohort of 18 children with recurrent or therapy-resistant germ cell tumours treated with high dose chemotherapy and subsequent autologous stem cell transplantation.
I have several minor comments to the manuscript:
Explanation of abbreviations of therapeutic regimen (VIP, ABK, BEP - no explanation in the text, HDCY) would improve the comprehensibility of the table. Adding information that patients 10-18 were treated with MEC 1 protocol and pts. 1-9 with MEC 2 would be beneficial.
Similarly in table 2 'age' should be specified (age of HDCT).
The analysis showed the age of HDCT as a significant prognostic factor. I would like to note that children of age 5-11 are missing in the analysed group. Ths fact is admitted by authors in discussion.
Reviewer 4 Report
This is a retrospective study comprising the “largest” reported pediatric cohort (n=18) of patients over a 15-year period with relapsing (9 patients) or therapy resistant (9 patients) extracranial germ cell tumors (10 M, 8F) after first-line chemo.
Median age at HDCT was 40 mos (17-222) with a bimodal distribution.
The 5-yr OS was 76%, EFS 70.8%. No therapy associated mortality. The 4 patients who died due to PD were 15.9, 17.7, 17.4, 14.7 yo.
Comments:
- The bimodal distribution (age<11 vs >11) reflects type I and II GCT. Perhaps, it is not surprising that 4/7 postpubertal patients died. Because they are distinct diseases, details of the pathology would be illuminating. For example, yolk sac tumor and teratoma are the most frequent tumors in the pediatric population. However, seminoma and embryonal carcinoma are rare. Yet, at least 5 patients had seminoma or embryonal carcinoma in this study cohort. (Unfortunately, 5 patients had unknown GCTs.) In Table 1, add column that shows whether patients have gonadal vs extragonadal disease (e.g., mediastinal or sacrococcygeal).
- Eight patients did not have metastatic disease. Comment on role of surgery to improve outcome, especially in the management of chemo-refractory nonseminomas. This information may account for local relapse in patients 1 & 16. Furthermore, residual disease in some patients may not be viable disease, in which case it is difficult to tell whether HDC actually contributed to improved outcome.
- Methods: Need to define primary resistant or relapsing within 6 months in Figure 1E. (Unfortunately, 2 patients had missing data on Rx). Similarly, need to clarify high-risk group due to high initial AFP (S2 and/or S3) or delayed response? (AFP T1/2 life>7 days). For example, it is unclear whether “not fulfilling CR criteria” means patients were still responding but not in favorably manner, so were their tumors primary resistant? Status at HDCT: 5 patients had CRs, all patients had PR, VGPR, or CR, nobody had PD. In other words, was second line or third line given for frank PD (n=9)? Otherwise for NSGCT, when markers normalize and residual tumors become resectable, shouldn’t those patients undergo surgery (n=9) rather than receive HDC+SCT?
- No difference in OS or EFS between MEC1 and MEC2 suggesting high-dose did not improve results. In fact, better agents or regimens may be important, similar to the goals of an ongoing international phase 3 “TIGER” trial (ref 11).
- The fact that 4 patients had PVB as 1st line treatment reminds us that this retrospective study was conducted over many years (2003-2018), which introduces additional selection biases as to the values of treatment vs experience in providing survival benefit in this rare disease.
- May wish to reference Nieto Y et al (Ann Oncol, 2015) regarding feasibility of mephalan in HDC+SCT for post-pubertal patients with GCT.
- Clarify whether embryonal ovarian carcinoma is different from embryonal carcinoma or is this transformation of embryonal carcinoma to ovarian carcinoma?
- Delete statement and reference regarding gonadal sex-cord stromal tumors in line 134, since this study focuses on GCT rather than on non-GCTs.
Round 2
Reviewer 2 Report
Many details are added to the manuscript, as requested, which is very much appreciated. These details help somewhat to understand the cohort and the treatment given to the patients. It remains, however, still unclear what exactly the indication for HDCT was in these patients. For some patients HDCT was applied as consolidation treatment within a second (or more) line treatment, for others as salvage treatment, although it is not even clear which patients had a really chemotherapy resistant active systemic disease at the moment of HDCT.
As childhood GCT is a heterogenous group of diseases, and because the prognosis of the most pediatric patients with a malignant GCT is very good, toxic and not proven treatment modalities should not be applied as standard treatment for these patients. Scientific evaluation of such an experimental treatment like HDCT is needed before it can be advised or proposed for particular situations. The retrospective analysis of such a small and heterogenous group is, however, not the right way to do this. In addition, the provided details are – for the most reported patients - insufficient to appreciate the exact clinical situation and thereby the possible impact of HDCT on the (individual) outcome. These principal rules and limitations make that the submitted description of the cohort has only minimal scientific merit.
The publication of this report in the current form could wrongly suggest for some readers that HDCT is an acceptable treatment modality for some pediatric patients with GCT, while no exact definitions of patient selection, indication and requirements are provided.
Sharing of these data could be justified but only with the right positioning and accompanied by a very critical reflection / discussion of the limitations emphasizing the complete lack of evidence of efficacy and the high toxicity of the regimen. A warning towards not using HDCT outside of a well-designed and controlled prospective clinical trial should also be placed.
Accordingly, the titel, discussion, conclusion and abstract should be revised.
Reviewer 4 Report
- Because refractory and lethal GCT in this pediatric patient population is rare, any reliable and pertinent data (see #11 below) could be invaluable for our knowledge base and patient care.
- However, when the sample size is small and the patient population is heterogeneous, this retrospective study may not be able to provide us reliable or pertinent data to inform us whether HDC+SCT is beneficial for the treatment of pediatric patients with refractory or relapsing GCT. It is also unclear how the data will guide us in the selection of patients and design of future studies for such treatment.
- For example, 4/7 postpubertal patients died, it seems that HDC+SCT may not be effective for this group of patients like their non-pediatric counterpart. However, none of the prepubertal patients died. Is this because of treatment or disease? Is HDC+SCT actual beneficial or even necessary? I don’t believe that this study is able to answer these questions.
- Therefore, separating the prepubertal and postpubertal groups will be illuminating, because they represent different diseases.
- In the era of chemotherapy, those postpubertal patients who die tend to succumb to chemo-resistant teratomas (Funt SA, J Clin Oncol 2016) and yolk sac tumors (Tu SM, Cancer 2016). Early/timely surgery may be more important than chemotherapy in the cure and care of these patients.
- It is notable that 6/11 prepubertal patients did not have metastatic disease at HDC+SCT. In Lines 70-71: “or presence of metastasis” were treated w/ HDC+SCT. Because patients 2, 4, 7, 8, 12, and 13 did not have metastasis at HDC+SCT, did they have active disease based on elevated AFP? If AFP level was normal, how could one be sure that the residual disease after chemo and surgery harbor viable (see #8 and 9 below) vs necrotic disease.
- Please, add another column depicting AFP level at HDC+SCT, since this is one of the putative high-risk (S3=AFP>10,000) factors for poor outcome in this patient population.
- Disseminated disease in lungs or LN (patients 3, 5, and 11) does not indicate high risk or implicate worse outcome for gonadal GCT.
- Importantly, 5/11 prepubertal patients did have metastatic disease at HDC+SCT, including 3 with non-pulmonary metastases. All five (patients 1, 10, 14, 16, and 17) had surgeries. Patients 1, 14, and 16 had viable disease according to pathology. Was this the basis for HDC+SCT? Were their AFP also elevated? Were residual metastatic disease at HDC+SCT actually viable in patients 10 and 17, because patient 17 received ABK after finding viable disease from his surgery but not before HDC+SCT?
- Pathology of those five patients (1, 10, 14, 16, and 17) who had surgeries prior to HDC+SCT could be informative. If teratoma, then more chemo even HDC+SCT (such as patient 1) may not be beneficial or necessary. (Hence, he had local relapse). If minimal residual disease (e.g., minute or focal viable tumor involving <10% or 2 cm of completely resected tumor), then most patients do not benefit or need additional treatment, either, because surgeries would have cured them (Fizazi K, J Clin Oncol 2001; Ann Oncol 2008).
- In the end, the crux of the data is in those 5 patients with metastatic extragonadal sacrococcygeal GCT (see #9 above) regarding the merits of HDC+SCT in prepubertal patients. The author could strengthen this study by providing a cohort of similar patients (i.e., metastatic extragonadal sacrococcygeal GCT) and demonstrating that those patients did just as well with standard chemo and surgery (or not) as the patients with HDC+SCT.
- The authors could mention that immunotherapy using anti-PD1 (Adra N, Ann Oncol 2018) and targeted therapy against the most common mutation, KIT, in GCT (Einhorn LH, Am J Clin Oncol 2006) have so far been disappointing.
Round 3
Reviewer 4 Report
I was disappointed that some critical information (to comments #7, 9, and 11) was not available or retrievable, because it would have greatly enhanced the value and merit of this paper. However, this is completely understandable, when we consider the rarity of disease and retrospective nature of the study.
I agree with the authors that this study contributes to our knowledge that “MEC is feasible in pediatric GCT, but associated with potentially life-threatening complications.”
I commend the authors’ valiant and honest efforts to obtain additional data to the best of their ability to help us face the challenge of patient complexity and solve tumor heterogeneity.
I recommend acceptance of this paper for publication, because the authors emphasize that “attempts to provide HDCT must be conducted according to well-designed clinical trials with the aim to identify those patients who may benefit from this approach to avoid unnecessary or overtreatment.”
Just to be consistent (with the ongoing TIGER trial), in the conclusions (lines 312-314): “The role of HDCT in GCT treatment is currently being evaluated and remains to be established in postpubertal patients. Whether this experience is applicable to pediatric prepubertal patients requires further study.”
Agree with additional editing of the amended manuscript before publication. “We are aware of many language errors and mistakes. The English language editing will be performed on the final draft of the manuscript.”
